# IMP3 Expression as a Potential Tumour Marker in High-Risk Localisations of Cutaneous Squamous Cell Carcinoma: IMP3 in Metastatic cSCC

**DOI:** 10.3390/cancers15164087

**Published:** 2023-08-14

**Authors:** Maurice Klein, Merle Wefers, Christian Hallermann, Henrike J. Fischer, Frank Hölzle, Kai Wermker

**Affiliations:** 1Department of Oral, Maxillofacial and Facial Plastic Surgery, School of Medicine, University Hospital RWTH Aachen, Pauwelsstrasse 30, 52074 Aachen, Germany; fhoelzle@ukaachen.de; 2Orthodontics Meyer, Kurze Straße 6, 48151 Muenster, Germany; merle-wefers@gmx.de; 3Laboratory for Dermatopathology and Pathology Hamburg-Niendorf, Tibarg 7, 22459 Hamburg, Germany; c.hallermann@drrm.de; 4Department of Dermatology and Histopathology, Fachklinik Hornheide, Dorbaumstrasse 300, 48157 Muenster, Germany; 5Department of Immunology, School of Medicine, University Hospital RWTH Aachen, Pauwelsstraße 30, 52074 Aachen, Germany; hefischer@ukaachen.de; 6Department of Oral and Cranio-Maxillofacial Surgery, Klinikum Osnabrueck GmbH, Am Finkenhuegel 1, 49076 Osnabrueck, Germany; kai.wermker@klinikum-os.de

**Keywords:** IMP3, lip cancer, squamous cell carcinoma, ear cancer, skin cancer, HNSCC

## Abstract

**Simple Summary:**

High IMP3 expression is correlated with poorer prognosis in many tumour entities. To date, there have been no data on IMP3 expression and clinical outcome in high-risk localisations (lip, ear) of squamous cell carcinoma of the skin. These are almost twice as likely to metastasise compared to other sites. In this study, the tumour marker IMP3 showed clear correlations with aggressiveness features (lymph node metastases, local recurrences, and progression-free survival). The identification of these more-aggressive tumours could influence therapy and diagnostics (radicality of neck dissection, follow-up intervals, staging). The analysis method presented here is efficient and could be easily incorporated into a clinical workflow and used for prospective testing.

**Abstract:**

Background: High IMP3 expression is correlated with a worse outcome. Until now, there have been no data about IMP3 expression and clinical outcome for high-risk localisation of squamous cell carcinoma of the skin (cSCC). Methods: One-hundred twenty-two patients with cSCC of the lip and ear were included, and IMP3 expression in the tumours was immunohistochemically assessed in different evaluation approaches. Subsequently, subgroups were analysed in a matched pair approach and correlated with clinical pathologic parameters. In the following, different IMP3 analysis methods were tested for clinical suitability. Results: We found a significant correlation between IMP3 expression and risk for lymph node metastasis, local relapse, and progression-free survival. Conclusions: On basis of our data, we suggest a prognostic benefit cutoff value for high (>50%) and low (<50%) IMP3 expression. Thus, IMP3 expression has a high scientific potential for further studies and could potentially be used as a prognostic marker in diagnostic and therapeutic decision-making.

## 1. Introduction

Squamous cell carcinoma of the skin (cSCC) is the second-most-frequent skin cancer after basal cell carcinoma [1]. Risk localisations of cSCC include the ear and lip, which display an increased risk of lymph node metastases (LNMs) compared to other tumour localisations. Alam et al. showed that the localisations of cSCC at the external ear (ECSCC) and lip (LSCC) with recurrence and metastasis rates of 8–25% are more aggressive than other localisations [2]. Furthermore, the localisation in the lip area has an influence on the tumour aggressiveness, so that LSCC is more aggressive when oral mucosa is affected [3]. However, the lymph node metastasis status (N+ or N−) is one of the most-important prognostic factors in cSCC of the head and neck [2,4,5]. The distinction between N+ and N− patients is very important, as N+ patients have a worse prognosis and a lower 5-year survival rate. N− patient had a 5-year survival of 87–95%, and N+ had only survival rates of 25–50% [6]. Especially patients with high-risk localisations LSCC and ECSCC benefit from risk prediction. In addition to the prognostic assessment, the risk evaluation of LNMs and the indication for neck dissection are controversial topics among different disciplines [1].

In terms of individualised medicine, there is a particular need for research on predictive tumour markers. For example, a predictive model for LSCC has been created. A study by Wermker et al. showed that, with the help of tumour thickness and grading, a risk stratification and evaluation for LNMs could be made [6]. In contrast to the histopathomorphological risk constellations, immunohistochemical markers could also be helpful in the prognostic evaluation of high-risk localisations of cSCC.

The insulin-like growth factor 2 mRNA-binding protein 3 (IGF2BP3, also named IMP3) could be a tumour marker with such a potential. IMP3 is an RNA-binding oncofetal protein [7]. Different studies have shown that these proteins have important implications in cell function, polarisation, cell migration, morphology, cellular metabolism, proliferation, and differentiation [7,8,9]. Gong et al. presented that IMP3 expression supports tumour cell proliferation, tumour cell adhesion, and tumour cell invasion [10]. There is also evidence of a link between increased IMP3 expression and advanced tumour stage [11]. In a meta-analysis in 2017, Chen et al. showed that the level of IMP3 expression correlates significantly with a decreased overall survival (OS) in different tumour entities. The authors evaluated 53 studies covering numerous tumour entities including renal cancer, lung cancer, oral cancer, and gastrointestinal cancer. There were positive correlations of high IMP3 expression with worse overall survival, disease-specific survival, and metastasis-free survival. To summarise, a high IMP3 expression is associated with a worse prognosis [12].

In oral squamous cell carcinoma (oSCC), high IMP3 expression correlates with lymph node metastasis (N+) and decreased 5-year survival. If two patients had the same tumour stage, but different IMP3 expression levels, the patient with the higher IMP3 expression had a worse prognosis [13].

The aim of our study was to analyse if the marker IMP3 can be used in a clinical setting to assess the aggressiveness of high-risk localisations of cSCC. The aggressiveness was determined with the overall survival rate, disease-specific survival, occurrence of local relapses, and progression-free survival. The key question was if the IMP3 analysis methods (IMP3 Analysis Category I (<25%, 25–50%, 50–75%, >75%), IMP3 Analysis Category II (0%, 1–20%, 21–60%, >60%), IMP3 Analysis Category III (>50%; <50%)) were usable for risk prediction for N+ cases (correlated with worse prognosis). It seems that one of the IMP3 analysis categories (IMP3 Analysis Category II) is particularly suitable in terms of outcome prediction.

## 2. Materials and Methods

### 2.1. Ethics Statement

This study was approved by the local ethics committee (Ethical Committee of the Westphalian Wilhelms University Muenster, Approval No. 2013-063-f-S) and was conducted in accordance with the Guidelines for Good Clinical Practice and in compliance with the Declaration of Helsinki. All patients gave their written informed consent for participating in this study.

### 2.2. Patients

All included patients were > 18 years old and had a histologically proven cSCC of the lip (LSCC) or ear (ECSCC). Included localisations for LSCC were the upper lip and lower lip. The following localisations were defined for the ECSCC: helix, cavum conchae/anthelix/tragus, retroauricular/posterior side, or a combination with more than one of these regions. All patients were resected R0 in the primary tumour.

Each patient had a preoperative stay with a minimum diagnostic procedure of sonography of the head and neck, X-ray of the thorax, and abdominal sonography. All patients were presented in an interdisciplinary tumour board. After the therapy, all patients received a periodic recall. The exclusion criteria were: no written consent and condition according to neck dissection or different HNSCC than in the inclusion criteria. Further data included in the correlation analysis were: follow-up time, first diagnosis, tumour localisation, local recurrence, lymph node metastasis, distant metastasis, disease-specific death, and overall survival.

The electronic patient data were complete and had a follow-up of at least 24 months. The abuse of alcohol or tobacco was not part of the exclusion criteria.

All patients (n = 122) were divided into two main groups and subsequently into two subgroups. Allocation to the two main groups was based on the localisation: LSCC and ESCC. These were then further subdivided into cases of N+ or N−. In addition, subgroup division into LSCC and ECSCC and lymph node status was performed: LSCC N−, LSCC N+, ECSCC N−, ECSCC N+.

In addition, a subdivision was performed for nodal status with both cases: N+ (LSCC N+ and ECSCC N+) and N− (LSCC N− + ECSCC N−).

The matched pair approach was used, as it allows a certain homogeneity to be achieved. The following groups were compared: LSCC N− vs. LSCC N+ and ECSCC N− vs. ECSCC N+. Each patient from the respective group was contrasted with a patient from the other group and with clinically pathologically selected known risk factors with as few differences as possible.

Included parameters for matching were: age, gender, grading, T-stadium, primary tumour localisation, tumour infiltration depth, perineural growth, cartilage invasion (only for ECSCC), comorbidities, and immunosuppression (classification into none, weak, moderate, strong). Finally, IMP3 expression (different IMP3 expression ranges; see below) was correlated with clinical pathological data.

### 2.3. Immunohistochemistry Analysis of IMP3

The selection of suitable tumour samples was made on Haematoxylin-Eosin-stained slices. The tumour tissue (histologically proven LSCC and ECSCC) was fixed in 10% formaldehyde solution.

Incubation with a primary antibody against IMP3 (anti-IMP3 Clon 69.1, 1:50, Agilent/Dako (Glostrup, Denmark)) was performed in the Autostainer Plus (Dako REAL DETECTION SYSTEM K5005, Glostrup). The tissue slices were incubated with the secondary (Dako REAL Link Biotinylated secondary antibody (AB2), Glostrup) antibody (exposure time 15 min) and then with Dako Real Streptavidin Alkaline Phosphatase (AP) (Glostrup; exposure time 15 min). Afterwards the SCC tissue slices were exposed to chromogen (Dako RED Chromogen, Glostrup) for 8 min. Next, Haematoxylin (nucleus counterstain/eight-minute exposure time) was applied. Coverslip tape was used.

The cellular localisation of IMP3 staining was determined by the Olympus BX51 microscope (Hamburg, Germany).

Two investigators blinded to the patients’ prognostic data analysed each slice with a first overview and then divided each slice into 5 high-power fields (HPFs) with a magnification of 400× after randomisation.

In every HPF, the investigator counted 5 × 10 tumour cells, totalling 50 cells per HPF and 250 cells in each slice. All tumour cells with a positive dark brown colour in the cytoplasm were counted as positive. Tonsil tissue was used as the positive control. The expression analysis was according to the literature [13,14]. In the event of discrepancies between the investigators, the case was reevaluated in a shared discussion.

The study focused on three different analysis categories. The aim was to find which expression range is most suitable for prognostic use. The expression was analysed in % ranges of expression: IMP3 Analysis Category I (<25%, 25–50%, 50–75%, >75%), IMP3 Analysis Category II (0%, 1–20%, 21–60%, >60%), IMP3 Analysis Category III (>50%; <50%). These three categories are presented in the Results Section.

### 2.4. Statistical Analysis

All statistical analyses were performed by using the Statistical Package for Social Sciences (SPSS) Version 22.0 for Windows^®^ (SPSS Inc., Chicago, IL, USA).

Categorical variables were analysed using the chi-squared test and Fisher’s exact test. For continuous variables, the Mann–Whitney U-test was used as a non-parametric test for abnormally distributed data, and an independent t-test was used to analyse normally distributed variables. Disease-specific survival (time from first diagnosis until tumour-dependent death; data on patients without tumour-dependent death were censored at the last follow-up time) and progression-free survival (time from first diagnosis until local recurrence or metastasising; data on patients without an event of progression were censored at the last follow-up time) were calculated using the Kaplan–Meier method, and group differences were analysed using the log-rank test.

## 3. Results

In the first part of the results, an overview of the patient cohort will be given, followed by the correlation of IMP3 expression with the nodal status groups in the second part. Finally, the evaluation methods of IMP3 expression (IMP3 Expression Categories I-III) are compared and correlated with clinical pathological outcome parameters.

### 3.1. Study Population

The patients’ age range was 42.7–97 years (mean 75.8 years, median 75.7 years, standard deviation 10.1 years). The patients’ subgroups were LSCC + ECSCC (n = 122, 100%, male n = 109, female n = 13). The subgroup data characteristics were: LSCC (n = 58, 47.5%, LSCC unilateral: n = 25; LSCC bilateral: n = 33) and ECSCC (n = 64, 52.5%).

The distribution of the classification of immunosuppression of the patients was as follows: none (n = 102), weak (n = 6), moderate (n = 13), strong (n = 1).

### 3.2. Clinical Pathology Data in the Study Cohort

Prognostic TNM data were collected: (all pT n = 122, 100%; pT1 n = 27, 22.5%; pT2 n = 31, 25.8%; pT3 n = 47, 37.5%; pT4 n = 17, 14.2%; all pN n = 120, 100%; pN0 n = 76, 63.3%; pN1 n = 26, 21.7%; pN 2a n = 14, 11.7%; pN 2b n = 3, 2.5%; pN 2c n = 1, 0.8%; all M n = 122, 100%; M0 n = 120, 98.4%; M1 n = 2, 1.6%). All patients showed an R0 status.

### 3.3. IMP3 Expression Distribution in the Study Cohort

As there were only marginal differences between the investigators, a shared decision was not necessary. The mean of IMP3 expression was 51.2%, and the median was 52.4% with a standard deviation of 19.5%. The range of IMP3 expression was between 0.0% and 88.4% IMP3 expression.

### 3.4. Higher IMP3 Expression Correlates with the Risk for Lymph Node Metastasis

The comparison between the N+ and N− groups showed that IMP3 expression in all cases with N+ had a mean of 61.4%, a median of 63.2%, and a standard deviation of 14.9.

Interestingly, IMP3 expression of all cases with N− showed a mean of 41.1%, a median of 45.6%, and a standard deviation of 18.4. Thus, a higher IMP 3 expression was correlated with pN+ in high-risk localisation of cSCC (*p* < 0.001). Figure 1 (left side) shows the correlation of the nodal status and IMP3 expression in the boxplot.

In addition to the correlation of the total collective (including both LSCC and ECSCC) with N+, the analysis of the subgroups LSCC and ECSCC followed. Interestingly, both subgroups showed a higher expression of IMP3 in the N+ group. Figure 1 (right side) displays the risk for LNMs and the expression of IMP3 in the subgroups of LSCC N+/N− and ECSCC N+/N− and IMP3 expression.

LSCC N+ patients presented with a mean IMP3 expression of 60.3%, a range of 0.0–85.2%, and a standard deviation of 16.2. For LSCC N−, we found a mean of 40.3%, a range of 0.0–72.4%, and a standard deviation of 17.9. Interestingly, the difference of LSCC N+ vs. LSCC N− was significant in the Mann–Whitney U-test in the subgroup analysis (*p* < 0.001), proving that higher IMP3 expression is correlated with the risk for LNMs in LSCC. This was also revealed in the analysis of the ECSCC N+ subgroup. These patients presented with a mean IMP3 expression of 62.3%, a median of 60.4%, a range of 32.4–88.4%, and a standard deviation of 13.8, whereas the ECSCC N− subgroup showed a mean of 41.8%, a median of 45.0%, a range of 0.0–75.2%, and a standard deviation of 19.1. The difference of ECSCC N+ v. ECSCC N− was also significant (*p* < 0.001) in the Mann–Whitney U-test. Taken together, IMP3 could be a reliable marker for metastasis risk assessment as it consistently correlated with the LNM rates.

### 3.5. IMP3 Correlates with Disease Progression and Local Relapse

In addition to the prediction of N+ cases by the IMP3 expression rates, other prognostically significant outcome parameters were also analysed: A higher IMP3 expression significantly correlated with disease progression (*p* < 0.001) and local relapse (*p* = 0.014). However, IMP3 expression did not correlate with disease-specific death (*p* = 0.090) and distant metastasis (*p* = 0.090).

### 3.6. IMP3 Analysis Categories I–III

After proving the prognostic potential of IMP3 expression analysis for pN+, a simple clinically applicable semiquantitative IMP3 expression analysis needed to be established. For this purpose, we analysed the IMP3 expression using three different approaches (Analysis Categories I–III) to identify the most-reliable and -application-oriented method for the prognostic evaluation of IMP3 expression in LSCC and ECSCC. In addition, it should be investigated whether the correlation between increased IMP3 expression and the increased occurrence of LNMs also applies to IMP3 Analysis Categories I–III.

### 3.7. IMP3 Analysis Category I (<25%, 25–50%, 50–75%, >75% IMP3 Expression)

For the first screening of IMP3 expression ranges, a quarter-step classification (IMP3 Analysis Category I; <25%, 25–50%, 50–75%, >75% IMP3 expression) was performed.

In IMP3 Category I, the age at first diagnosis was as follows: <25% IMP3 expression: mean 76.1 years, median 73.6, 64.7–93.9 years; 25–50% IMP3 expression: mean 76.6 years, median 77.9 years, 42.7–94.8 years; 50–75% IMP3 expression: mean 74.9, median 75.2 years, 44.3–97.0 years; >75% IMP3 expression: mean 76.8 years, median 76.1 years, 65.2–91.7 years. The correlation was not significant in the Pearson chi-squared test (*p* = 0.346).

A distribution of IMP3 expression for the AJCC was also shown: <25% IMP3 expression: Stage I (n = 5), Stage II (n = 3), Stage III (n = 1), Stage IV (n = 3); 25–50% IMP3 expression: Stage I (n = 10), Stage II (n = 8), Stage III (n = 19), Stage IV (n = 9); 50–75% IMP3 expression: Stage I (n = 7), Stage II (n = 8), Stage III (n = 24), Stage IV (n = 14); >75% IMP3 expression: Stage I (n = 1), Stage II (n = 0), Stage III (n = 7), Stage IV (n = 3). The correlation failed to reach significance in the Pearson chi-squared test (*p* = 0.179).

In addition, IMP3 Analysis Category I is presented with the degree of differentiation (grading): <25% IMP3 expression: G1 (n = 4), G2 (n = 6), G 3 (n = 2); 25–50% IMP3 expression: G1 (n = 10), G2 (n = 27), G 3 (n = 9); 50–75% IMP3 expression: G1 (n = 11), G2 (n = 32), G 3 (n = 10); >75% IMP3 expression: G1 (n = 1), G2 (n = 8), G 3 (n = 3). The correlation was not significant in the Pearson chi-squared test (*p* = 0.905).

The distribution of the strength of immunosuppression could also be shown as dependent on the IMP3 expression: <25% IMP3 expression: none (n = 11), weak (n = 1), moderate (n = 0), strong (n = 0); 25–50% IMP3 expression: none (n = 38), weak (n = 4), moderate (n = 4), strong (n = 0); 50–75% IMP3 expression: none (n = 42), weak (n = 1), moderate (n = 9), strong n = 1); >75% IMP3 expression: no patient with immunosuppression. The correlation failed to reach significance in the Pearson chi-squared test (*p* = 0.381).

There were more cases in the N+ (LSCC N+ and ECSCC N+) group with an IMP3 expression range of 50–75% (63.9%) and an IMP3 expression range >75% (16.4%) as compared to the respective N− groups.

In line with this, the quartiles with lower IMP3 expression ranges showed fewer cases in the N+ (LSCC N+ and ECSCC N+) group (IMP3 expression range <25% (1.6%) and IMP3 expression range of 25–50% (18.0%)).

As expected, there were significantly more cases in the N− (LSCC N− + ECSCC N−) group with an IMP3 expression range <25% (18.0%) or IMP3 expression range of 25–50% (57.4%) and fewer cases for the high expression ranges (IMP3 expression range of 50–75% (23.0%) and IMP3 expression range >75% (1.6%)).

The cross-tabulation showed significance (*p* < 0.001) for IMP3 Analysis Category I with nodal status in the Pearson chi-squared test.

We then analysed this in further detail by comparing the individual subgroups (LSCC N−, LSCC N+, ECSCC N−, ECSCC N+) and found that, generally, the lower quartiles were more frequent in the N− subgroups, whereas in the N+ subgroups, the higher quartiles were prevalent: LSCC N−: IMP3 expression <25% (17.2%), IMP3 expression of 25–50% (65.5%), IMP3 expression of 50–75% (17.2%), IMP3 expression >75% (0%); LSCC N+: IMP3 expression <25% (3.4%), IMP3 expression of 25–50% (13.8%), IMP3 expression of 50–75% (75.9%), IMP3 expression >75% (6.9%); ECSCC N−: IMP3 expression <25% (18.8%), IMP3 expression of 25–50% (50.0%), IMP3 expression of 50–75% (28.1%), IMP3 expression >75% (3.1%); ECSCC N+: IMP3 expression <25% (0%), IMP3 expression of 25–50% (21.9%), IMP3 expression of 50–75% (53.1%), IMP3 expression >75% (25.0%).

After subgroup analysis, the correlations of IMP3 Analysis Category I with different clinical pathological outcome parameters were examined for all patients.

A significant correlation between IMP3 expression and N+ (*p* < 0.001) was demonstrated for IMP3 Analysis Category I.

There was no significant correlation (*p* = 0.370) of IMP3 Analysis Category I and disease-specific death in the Pearson chi-squared test. In contrast, there was a correlation in IMP3 Analysis Category I and disease progression (*p* < 0.001). In addition, trends could be shown for IMP3 Analysis Category I and distant metastasis (*p* = 0.093) and IMP3 Analysis Category I and local relapse (*p* = 0.058).

As we saw before that IMP3 expression correlates with disease-free progression, we also performed a Kaplan–Meyer analysis for progression-free survival for IMP3 Analysis Category I: Figure 2 shows that progression-free survival was reduced for an IMP3 expression range > 50%. The log-rank test (comparison with the <25% IMP3 expression range) showed a significant difference with the 50–75% IMP3 expression range (*p* = 0.007) and >75% IMP3 expression range (*p* = 0.004). For the 25–50% IMP expression range, no significant difference could be shown.

### 3.8. IMP3 Analysis Category II (0%, 1–20%, 21–60%, >60% IMP3 Expression)

After the analysis of the IMP3 Analysis Category I (<25%, 25–50%, 50–75%, >75% IMP3 expression), the IMP3 expression was subdivided into thirds (IMP3 Analysis Category II; 0%, 1–20%, 21–60%, >60% IMP3 expression).

In IMP3 Category II, the age at first diagnosis was as follows: 0% IMP3 expression: mean 69.5 years, median 71.6 years, 64.7–72.1 years; 1–20% IMP3 expression: mean 78.8 years, median 77.4 years, 69.5–93.9 years; 21–60% IMP3 expression: mean 75.8 years, median 76.5 years, 42.7–97.0 years; >60% IMP3 expression: mean 75.8 years, median 76.0 years, 44.3–97.0 years. The correlation was not significant in the Pearson chi-squared test (*p* = 0.223).

In addition, a distribution of IMP3 Analysis Category II could be shown for the AJCC: 0% IMP3 expression: Stage I (n = 2), Stage II (n = 1), Stage III (n = 0), Stage IV (n = 0); 1–20% IMP3 expression: Stage I (n = 2), Stage II (n = 2), Stage III (n = 1), Stage IV (n = 3); 21–60% IMP3 expression: Stage I (n = 13), Stage II (n = 15), Stage III (n = 29), Stage IV (n = 14); >60% IMP3 expression: Stage I (n = 6), Stage II (n = 1), Stage III (n = 21), Stage IV (n = 12). The correlation was significant in the Pearson chi-squared test (*p* = 0.042).

In addition, the IMP3 Analysis Category I is presented with the degree of differentiation (grading): 0% IMP3 expression: G1 (n = 2), G2 (n = 1), G 3 (n = 0); 1–20% IMP3 expression: G1 (n = 2), G2 (n = 4), G 3 (n = 2); 21–60% IMP3 expression: G1 (n = 16), G2 (n = 42), G 3 (n = 13); >60% IMP3 expression: G1 (n = 6), G2 (n = 26), G 3 (n = 8). The correlation was not significant in the Pearson chi-squared test (*p* = 0.522).

The distribution of the strength of immunosuppression could also be shown as a dependent on the IMP3 expression: 0% IMP3 expression: no immunosuppression (n = 3); 1–20% IMP3 expression: none (n = 7), weak (n = 1), moderate (n = 0), strong (n = 0); 21–60% IMP3 expression: none (n = 58), weak (n = 4), moderate (n = 8), strong (n = 1); >60% IMP3 expression: none (n = 34), weak (n = 1), moderate (n = 5), strong (n = 0). The correlation failed to reach significance in the Pearson chi-squared test (*p* = 0.922).

In the N+ group (LSCC N+ + ECSCC N+), most patients (54.1%) were in the >60% IMP3 expression range and 44.3% in the 21–60% IMP3 expression range. In the 0% IMP3 expression range, there were 1.6% of the cases, and in the 1–20% IMP3 expression range, no patients (0%).

The distribution in the N− group (LSCC N− + ECSCC N−) was different. Here, in the IMP3 expression range group 0%, 3.3% of the cases, in the IMP3 expression range 1–20%, 13.1% of the cases, in the IMP3 expression range 21–60%, 72.1% of the cases, and in the IMP3 expression range >60%, 11.5% of the cases fell.

IMP3 expression in the subgroups (LSCC N−, LSCC N+, ECSCC N−, ECSCC N+) was also examined:

LSCC N−: IMP3 expression range of 0%: 3.4%, IMP3 expression range of 1–20%: 10.3%, IMP3 expression range of 21–60%: 79.3%, IMP3 expression range >60%: 6.9%; LSCC N+: IMP3 expression range of 0%: 3.4%, IMP3 expression range of 1–20%: 0%, IMP3 expression range of 21–60%: 37.9%, IMP3 expression range >60%: 58.6%; ECSCC N−: IMP3 expression range of 0%: 3.1%, IMP3 expression range of 1–20%: 15.6%, IMP3 expression range of 21–60%: 65.6%, IMP3 expression range >60%: 15.6%, ECSCC N+: IMP3 expression range of 0%: 0%, IMP3 expression range of 1–20%: 0%, IMP3 expression range of 21–60%: 50.0%, IMP3 expression range >60%: 50.0%.

No correlation was found between IMP3 Analysis Category II and the risk of disease- related death and local relapse.

In contrast, the Pearson chi-squared test showed a positive correlation of IMP3 Analysis Category II with disease progression (*p* = 0.001), distant metastasis (*p* = 0.014), and LNM (*p* = 0.008).

### 3.9. IMP3 Analysis Category III (<50%, >50% IMP3 Expression)

A quick and clinically easy way to implement IMP3 analysis is the classification into the >50% and <50% IMP3 expression ranges (IMP3 Analysis Category III).

In IMP3 Category III, the age at first diagnosis was as follows: >50% IMP3 expression: mean 75.3 years, median 75.7 years, 44.3–97.0 years; <50% IMP3 expression: mean 76.5 years, median 76.1 years, 42.7–94.8 years. The correlation was not significant in the Pearson chi-squared test (*p* = 0.397).

A distribution of IMP3 expression for the AJCC was also shown: >50% IMP3 expression: Stage I (n = 8), Stage II (n = 9), Stage III (n = 33), Stage IV (n = 18); <50% IMP3 expression: Stage I (n = 15), Stage II (n = 10), Stage III (n = 18), Stage IV (n = 11). The correlation failed to reach significance in the Pearson chi-squared test (*p* = 0.80).

In addition, the IMP3 Analysis Category I is presented with the degree of differentiation (grading): >50% IMP3 expression: G1 (n = 12) G2 (n = 43) G 3 (n = 13); <50% IMP3 expression: G1 (n = 14) G2 (n = 30) G 3 (n = 10). The correlation was not significant in the Pearson chi-squared test (*p* = 0.530).

The distribution of the strength of immunosuppression could also be shown as dependent on the IMP3 expression: >50% IMP3 expression: none (n = 57), weak (n = 1) moderate (n = 9) strong (n = 1); <50% IMP3 expression: none (n = 45), weak (n = 5) moderate (n = 4) strong (n = 0). The correlation failed to reach significance in the Pearson chi-squared test (*p* = 0.141).

The >50% IMP3 expression range was positively correlated with the presence of LNMs at all sites (LSCC N+ + ECSCC N+): 83.6% of these cases showed an LNM and 27.9% no LNM. In contrast, 72.1% of patients in the <50% IMP 3 expression range group did not have an LNM (LSCC N− + ECSCC N−), and 16.4% had an LNM. Thus, a higher IMP3 expression of >50% was significantly (*p* < 0.001) correlated with the occurrence of an LNM.

We next analysed the localisation subgroups and also found significant correlations with IMP3 Expression Category III. In the LSCC group and IMP3 expression >50%, 82.8% of patients had an LNM and only 24.1% had no LNM. In the LSCC group and IMP3 expression <50%, 75.9% of patient cases showed no LNM and only 17.2% had LNM.

Comparable results were also found in the ECSCC group. In the ECSCC group and an IMP3 expression of >50%, 84.4% of patients had an LNM and 31.3% had no LNM. In line with this, in the ESCC group and an IMP3 expression of <50%, 68.8% of the patients had no LNM and 15.6% had an LNM. IMP3 Expression Category III (<50%, >50% IMP3 expression) was significantly correlated (*p* < 0.001) with the risk for LNMs in the subgroups.

Although we were able to show the significant correlation of >50% IMP3 expression and LNMs, we could unfortunately not demonstrate a significant correlation for IMP3 expression and distant metastasis in this category. Furthermore, there was a correlation between IMP3 Analysis Category III and disease progression (*p* < 0.001).

In addition, it was investigated whether IMP3 Analysis Category III correlated with the risk of local recurrence. There was a correlation (*p* = 0.012) for IMP3 Analysis Category III and local recurrence in the Pearson chi-squared test.

To examine whether the 50% expression cutoff was also able to predict the disease-specific survival, we analysed IMP3 Expression Category III with disease-related death. Higher IMP3 expression showed a tendency (*p* = 0.092) towards disease-related death in the IMP3 expression category in the log rank test (Mantel–Cox). A higher IMP3 expression of >50% showed a lower disease-specific survival (Figure 3).

Another important clinical outcome parameter is progression-free survival. A higher IMP3 expression >50% was correlated also with a shorter progression-free survival. The log rank test (Mantel–Cox) showed significant differences (*p* < 0.001). Figure 4 demonstrates the Kaplan–Meier curve for progression-free survival and IMP3 expression (IMP3 Analysis Category III (<50%, >50%)).

### 3.10. Concluding Remarks on the Results

The tumour marker IMP3 seems to be suitable for outcome prediction in LSCC and ECSCC. We observed a significant risk prediction potential for LNMs and, in particular, with IMP3 Analysis Category III (<50%, >50%). The evaluation was fast, efficient, and simple.

## 4. Discussion

### 4.1. Strengths and Weaknesses of the Study

The strength of the study is a high number of rare cases of cSCC of the ear and lip with lymph node metastasis. However, because of the distribution of the groups of N+ and N−, a selection bias is possible. Nevertheless, with the matched pair approach, it is possible to minimise the effects of known risk factors for bad prognosis and to work out the effect of the marker IMP3 in small cohorts. The matched pair approach nevertheless triggers a selection bias, which is why a multivariate analysis was deliberately omitted.

A weakness of retrospective analysis in general is that the data quality is lower than in a prospective approach, but for preclinical testing of IMP3, it is still more efficient than a prospective approach.

The classification of the expression ranges presented here can also be critically questioned. However, there is no standardised evaluation method for IMP3 in the literature, and different cutoff values have been published for when expression is considered positive [12]. It should be added that the evaluations also vary with regard to the semi-quantitative analysis.

Another strength of our study is that the IMP3 expression analysis was consistent between the two investigators, indicating that the analysis method seems to be reliable and independent of the examiner.

The origin of LSCC is often not detectable. It is possible that the origin is the oral mucosa, the white of the lip, or the red of the lip. This makes the exact onset of the disease and, thus, the precise classification into oSCC, cSCC, or LSCC difficult [15]. Unfortunately, this could have an impact on tumour aggressiveness. Oral SCCs are much more aggressive than SCCs of the skin [3]. The LSCC shows an intermediary status in aggressiveness. This is the reason why the LSCC is more and more seen as an independent tumour entity. In addition, the outer lip was found to be about 40-times more frequently affected by LSCC and other tumour entities than the inner lip [16]. Our study included upper lip and lower lip cancers because many studies did not show a prognostic difference between upper lip and lower lip cancers [1,17,18,19].

In the present study, the patient’s phototype was not recorded. The phototype could also have an influence on the outcome in connection with IMP3 expression.

However, this is an academic discussion, and for the clinician, the assessment of aggressiveness is often, nonetheless, difficult. It might be reasonably assumed that lymph drainage is not the same in the comparison of the lip and ear. Likewise, the tumour biology is perhaps different. Nevertheless, we strongly believe that the strengths of our study outweigh the limitations.

### 4.2. Is IMP3 Expression Useful for the Clinical Outcome Prediction of High-Risk Localisation in cSCC?

In the literature, IMP3 is examined in many different tumour entities [20,21,22].

The RNA-binding protein IMP3 is known as a cancer-specific gene [10,12]. An increased IMP3 expression was observed in both malignant and benign neoplasm [7,12]. Furthermore, it is possible to differentiate between cSCC and keratoacanthoma with the marker IMP3 [23]. Chen et al. showed that an increased IMP3 expression is associated with a high recurrence and metastasising rates in many tumour entities [12]. This predisposes the protein IMP3 as a tumour marker for oncology decision-making. Many therapeutic decisions such as radiation or radio-chemotherapy are dependent on lymph node status. Liao et al. showed that IMP3 promotes the proliferation, cell growth, and robustness against ionising radiation [24]. Thus, one can speculate that IMP3 can be used to assess the response rate or indication for the radiotherapy of cSCC.

Another useful application of IMP3 could be the outcome prediction of cSCC during follow-up. In our study, we investigated this potential and aimed to identify the best assessment method. We found that a high IMP3 expression >50% showed a shortened progression-free survival into the seventh year. Especially in the first 24 months, the progression-free survival deteriorates strongly with an IMP3 expression >50%. After approximately 24–36 months, there is no or only marginal change in progression-free survival. Our data suggest that patients with an IMP3 expression >50% should be followed up more closely because of higher event risk. The marker IMP3 could thus be the basis for the decision of after-care duration of high-risk cSCC patients. However, this hypothesis must be controlled by bigger prospective cohorts.

In addition, we found a tendency (but a lack of significance) towards poorer disease- specific survival with increased IMP3 expression >50%. It seems sensible and exciting to investigate this connection in a larger collective.

It is known that IMP3 expression is different in the main tumour or satellite cells [13]. Future studies should also evaluate outcome differences when IMP3 expression is studied in the primarius tumours or in the LNMs.

Our study found no significant correlation between distant metastasis and IMP3 expression. However, this might be due to the small number of patients with distant metastasis. An increased risk for distant metastasis is permissible because of the correlation of lymph node metastasis and distant metastasis [25]. This fact must be controlled for IMP3 by bigger cohorts.

Taken together, the IMP3 expression is a promising candidate for risk assessment of high-risk cSCC in the clinic. Furthermore, it has a high scientific potential for further studies and could potentially be used as a prognostic marker in diagnostic and therapeutic decision-making.

### 4.3. Which Is the Best IMP3 Expression Category for Clinical IMP3 Outcome Prediction?

After confirming the beneficial outcome prediction of IMP3 in high-risk cSCC, the question of IMP3 expression range analysis (IMP3 analysis categories) should be discussed in terms of application areas and potential clinical use. Different clinical endpoints can be defined for outcome prediction. Lin et al. showed a correlation between increased IMP3 expression and a decreased 5-year survival [13]. Our data showed that IMP3 Analysis Category III (<50%, >50%) could be used for follow-up assessment and prediction of progression-free survival. A shorter progression-free survival, especially in the first months, was correlated with IMP3 expression >50%. A new approach could be to adjust tumour follow-up care dependent on IMP3 expression. Since the risk of local relapse is increased with high IMP3 expression, this could support the call for IMP3-dependent closely monitored follow-up care.

In addition, the nodal status is important for the discussion of staging and therapy. The finding of significant differences in group and subgroup analysis made it necessary to find a suitable analysing system and cutoff value for the evaluation of potential clinical use for practitioners. The higher expression of IMP3 in the N+ group of LSCC and ECSCC requires the search for a cutoff value to use IMP3 as an LNM prediction marker. For this also, it seems that IMP3 Analysis Category III (>50%; <50%) is best suited. At an IMP3 expression >50%, the risk for LNMs increased. A similar correlation of IMP3 expression and the risk for LNMs was found in literature for oral cancer [13]. The few categories are advantageous for quick and cost-effective implementation for the clinician for prognostic prediction of LNMs. The discrimination of the expression level into just these two ranges is till sufficient for risk assessment and, then, is the easiest, yet still reliable method for clinical use.

### 4.4. IMP3 Expression Analysis Could Potentially Help in Decision-Making of Neck Dissection

There is wide knowledge about many tumour entities with a correlation between IMP3 and the risk for LNMs or DMs [12,13,26,27,28]. For oSCC and also for squamous cell carcinoma of the uterine cervix, a significant correlation between IMP3 and LNMs has been shown [13,26]. In contrast, in another tumour entity, prostate carcinoma, increased IMP3 expression of the primary tumour was shown in the presence of distant metastasis [28].

This correlation shows that IMP3 is unfortunately not specific to cSCC and can also be found in other tumour entities. Nevertheless, it also shows that IMP3 seems to play a role in many tumour entities with regard to metastasis.

Thus, we aimed to analyse if the IMP3 expression status is usable for LNM risk evaluation in high-risk localisations of cSCC. The LNM is an important clinical tool to estimate the prognosis. There is always a discussion about the extent of neck dissection in LSCC and ECSCC. The therapeutic consequence is often a neck dissection, but for the patient, it has a huge impact on the quality life. The decision of a neck dissection often depends on the orientation of a conservative or more-surgical medical discipline. IMP3 could help to support the decision for or against neck dissection in the high-risk localisations of cSCC.

The next question could be as follows: Should every patient with cSCC get an IMP3 expression status? Our opinion is that only the patients with high-risk localisation of the lip and ear benefit from this diagnostic tool because of the increased risk for LNMs. Without randomised, multi-centre, and controlled studies, there is no final statement for this biomarker. Nevertheless, there is a good cost–benefit for the reproducible immunohistochemical marker IMP3. More studies are needed to investigate this promising and exciting tumour marker in cSCC.

### 4.5. Outlook on IMP3 Vaccination/IMP3 Therapy

In advanced oesophageal cancer, IMP3 and other peptides (TTK, LY6K) have been used therapeutically as vaccines in phase II clinical trials. The authors postulate that an improvement in prognosis could be achieved [29]. Similarly, prognosis improvement by vaccination with IMP3 and other peptides (LY6K, CDCA1) has been used in advanced HNSCC in phase II clinical trials [30]. However, it remains to be seen whether these therapies will become established and whether they can find their way into the treatment of high-risk localisations of cSCC. Unfortunately, there is currently no therapy available that targets IMP3. However, due to the frequent occurrence of IMP3 expression, especially in aggressive tumours, the target structure appears to be interesting for system therapies. Interestingly, there is some evidence that IMP3 could be used to assess chemosensitivity in triple-negative breast cancers [31,32]. For example, Ohashi et al. showed that IMP3-positive tumours were significantly more likely to be non-responders to neoadjuvant chemotherapy [32]. It remains to be seen whether an influence of IMP3 on chemosensitivity can also be shown for cSCC.

## 5. Conclusion

In summary, the immunohistochemical marker IMP3 is suitable as a prognostic marker in high-risk localisation of cSCC. All three applied analysis categories showed significances in their correlation with the clinical outcome of the patients. A classification into <50% and >50% expression seems to be easily applicable, reproducible, and efficient and, thus, is the most-promising strategy to apply in the clinic. Furthermore, IMP3 assessment could help in the decision-making of radical neck dissection or could reduce non-indicated neck dissection in high-risk cSCC. Lastly, IMP3 expression can also be used to identify aggressive tumours early on and to adjust the patients’ follow-up care.

## Figures and Tables

**Figure 1 cancers-15-04087-f001:**
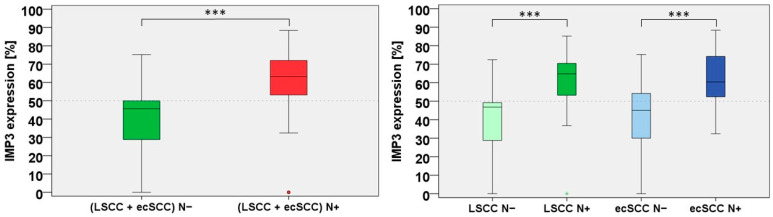
(**Left side**) IMP3 expression correlates positively with nodal status (N+/N−) in high-risk localisation of squamous cell carcinoma of the ear (ECSCC) and lip (LSCC) (*p* < 0.001). (**Right side**) The subgroups analysis of LSCC N+/N− and ECSCC N+/N− and IMP3 expression show each a significant correlation with nodal status (*p* < 0.001). *** *p* < 0.001.

**Figure 2 cancers-15-04087-f002:**
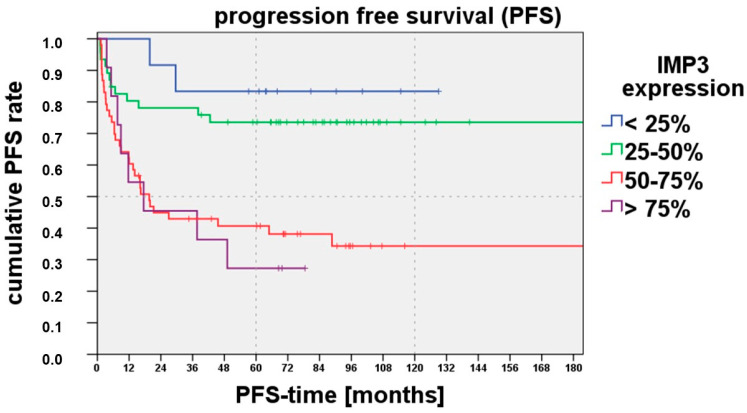
The progression-free survival was shortened depending on the IMP3 expression ranges (IMP3 Analysis Category I; <25%, 25–50%, 50–75%, >75%). A higher IMP3 expression of >50% was correlated with a less progression-free survival.

**Figure 3 cancers-15-04087-f003:**
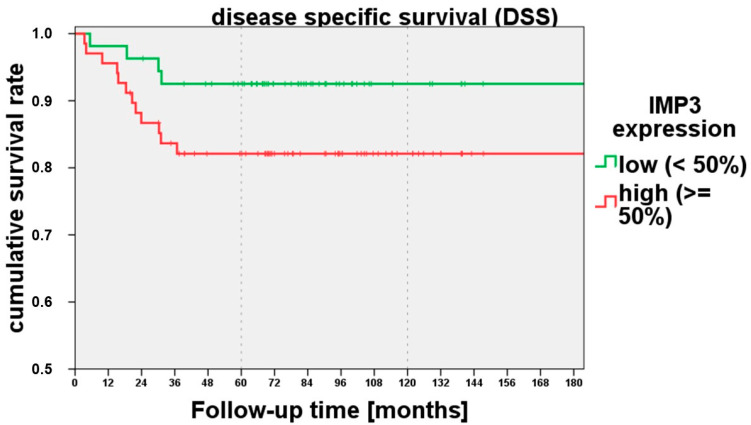
The disease-specific survival tended to be reduced (*p* = 0.092) from an IMP3 expression of >50% (IMP3 Analysis Category II).

**Figure 4 cancers-15-04087-f004:**
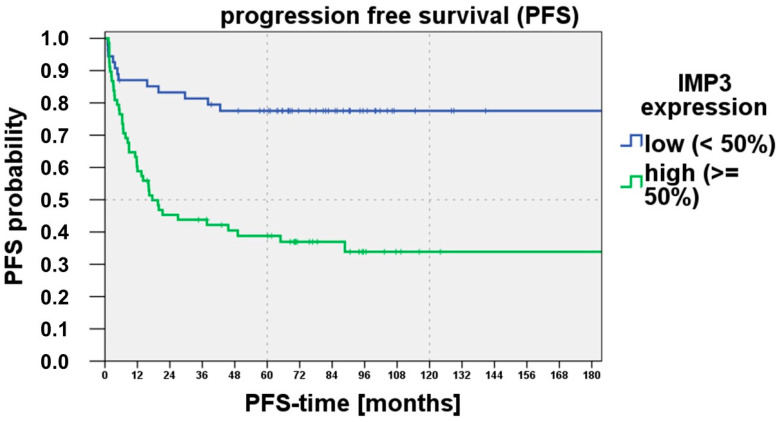
The IMP3 expression range >50%, <50% (IMP3 Analysis Category II) showed a significantly (*p* < 0.001) reduced progression-free survival with an IMP3 expression of >50%.

## Data Availability

The research data can be requested after consultation with the corresponding author.

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
