# Peer review of "IMP3 Expression as a Potential Tumour Marker in High-Risk Localisations of Cutaneous Squamous Cell Carcinoma: IMP3 in Metastatic cSCC"

_cancers, 2023, doi:10.3390/cancers15164087_

Round 1
Reviewer 1 Report
Klein et al highlighted the correlation of the expression of IMP3 and the risk for lymph node metastasis, local relapse, and poor prognosis, and proved IMP3 can be a potential tumor marker in high-risk localizations of cutaneous squamous cell carcinoma. Overall, the study is well performed, but there are some minor points the author would need to address.
Minor points:
1. It will be great if the author could discuss more is IMP3 specific high expressed only in cutaneous squamous cell carcinoma or it can also be found in other type of metastasis tumor.
2. Need more information or discussion about is there any inhibitor or treatment to target on IMP3.
3. The figure labels need to be clearer.
Author Response
Response to the reviewers:
We would like to thank all reviewers for their helpful comments and constructive criticism. We are confident that they greatly helped to improve the manuscript. We have addressed all questions and concerns as indicated below. All changes in the manuscript are marked green in the text.
Reviewer 1
Klein et al highlighted the correlation of the expression of IMP3 and the risk for lymph node metastasis, local relapse, and poor prognosis, and proved IMP3 can be a potential tumor marker in high-risk localizations of cutaneous squamous cell carcinoma. Overall, the study is well performed, but there are some minor points the author would need to address.
Minor points:
- It will be great if the author could discuss more is IMP3 specific high expressed only in cutaneous squamous cell carcinoma or it can also be found in other type of metastasis tumor.
Reply: Thank you very much for the comment. We added in the discussion that IMP3 is found in many metastatic tumor entities. IMP3 is unfortunately not specific for cutaneous squamous cell carcinomas (p. 26; discussion).
- Need more information or discussion about is there any inhibitor or treatment to target on IMP3.
Reply: Unfortunately, up to now there is no therapy available that targets IMP3. However, due to the frequent occurrence of IMP3 expression, especially in aggressive tumors, the target structure appears to be interesting for system therapies (p. 27; discussion).
- The figure labels need to be clearer.
Reply: We have added the figures in an improved resolution (p. 12, 16, 21; results).
Reviewer 2 Report
Klein et al. evaluated the IMP3 immunohistochemical expresion in 122 cases of squamous cell carcinoma (SCC) of the lip and the ear, with the assumption that the IMP3 expression is an unfavourable prognostic factor in several other human neoplasms and that the lips and the ear are high-risk locations for cutaneous SCC. They found that IMP3 expression wass much commoner in node-positive cases of SCC compared with node-negatives (median values: 63.2+/-14.9% vs 45.6+/-18.4%); moreover the IMP3 expression correlated with disease progression and local relapse. The Authors also grouped the IMP3 expression into 3 different categories (I: <25% vs 25-50% vs 50.75% vs >75%; II: 0% vs 1-20% vs 21-60% vs >60%; III (>50% vs <50%). Thus they found that the best discriminating value for prognostic assessment was 50% (category III).
Besides some inconsistencies in the categorization (in category I, 50% is shared in two different groups; in category II a better grouping might be 0% vs 1-33% vs 34-66% vs >66%; in category III 50% is apparently not counted), I think that the manuscript does not give any clinically valuable information, since a mutivariate anaysis was not performed. The different categories of IMP3 expression should be indeed matched at least with clinica/pathologic stage, grade, modified BT, and status of the surgical margins; ideally, also age and phototype of patients should be considered; a state of immune suppression should be finally annotated.
Finally, the immunohistochemical technique should be not itemized in detail.
Author Response
Response to the reviewers:
We would like to thank all reviewers for their helpful comments and constructive criticism. We are confident that they greatly helped to improve the manuscript. We have addressed all questions and concerns as indicated below. All changes in the manuscript are marked green in the text.
Reviewer 2
Klein et al. evaluated the IMP3 immunohistochemical expression in 122 cases of squamous cell carcinoma (SCC) of the lip and the ear, with the assumption that the IMP3 expression is an unfavorable prognostic factor in several other human neoplasms and that the lips and the ear are high-risk locations for cutaneous SCC. They found that IMP3 expression was much commoner in node-positive cases of SCC compared with node-negatives (median values: 63.2+/-14.9% vs 45.6+/-18.4%); moreover the IMP3 expression correlated with disease progression and local relapse. The Authors also grouped the IMP3 expression into 3 different categories (I: <25% vs 25-50% vs 50.75% vs >75%; II: 0% vs 1-20% vs 21-60% vs >60%; III (>50% vs <50%). Thus they found that the best discriminating value for prognostic assessment was 50% (category III).
Besides some inconsistencies in the categorization (in category I, 50% is shared in two different groups; in category II a better grouping might be 0% vs 1-33% vs 34-66% vs >66%; in category III 50% is apparently not counted), I think that the manuscript does not give any clinically valuable information, since a multivariate analysis was not performed.
Reply: We thank you for the valuable suggestions. A division into "0% vs 1-33% vs 34-66% vs >66%" could have certainly been made. We made our decision for the evaluation method based on other studies: In the meta-analysis by Chen et. al (DOI: 10.2147/OTT.S128810, pages 2852-2853) it is very clear that a wide variety of analysis methods were used in different tumor entities between 2006 - 2016. The cut-off values vary greatly in the literature. A standardized cut-off value and a semi-quantitative IMP3 expression classification do not exist. We have included your argument in the discussion of strengths and weaknesses (p. 22; discussion).
Furthermore, thank you for your suggestion to perform multivariate analysis. We discussed this item with our statistician. Due to our study design and the patient selection (matched pairs) there is always a selection bias compared to the parent population. This means that due to our matched-pairs-study-design, all relevant clinicopathological parameters were similar in both the metastasis and the non-metastasis group. The advantage of this approach is the control of confounders, however, the disadvantage is a loss of similarity to the underlying population. E.g., there are much more metastasis-cases compared to a consecutive cohort without matched pairs. Thus, it is not suitable to perform multivariate analyses in our study because all other variables were similar in the groups. We have noted this connection in the discussion section under strengths and weaknesses (p. 22 discussion).
The different categories of IMP3 expression should be indeed matched at least with clinical/pathologic stage, grade, modified BT, and status of the surgical margins; ideally, also age and phototype of patients should be considered; a state of immune suppression should be finally annotated.
Reply: Thank you very much for the comments. All patients had a R0 status. We have added this fact in the methods and results part (p. 7 methods, 10 results). In addition, we now included data for age, clinical pathological stage, grading and immunosuppression discuss them for the respective IMP3 analysis categories (p. 13, 14, 16, 17, 18, 19; results). Furthermore, we classified the immunosuppression into none, weak, moderate, and strong in the methods (p. 8) and results part (p.10). Unfortunately, the patient's phototype was not collected in the data and we included this point of criticism in the discussion section strength and weaknesses (p. 22; discussion).
Finally, the immunohistochemical technique should be not itemized in detail.
Reply: Thank you for this suggestion. We followed your advice and have shortened the explanation of the immunohistochemical staining to the most essential (s. 8 methods).
Round 2
Reviewer 2 Report
None